# Exploring the Mechanisms Underlying the Cardiotoxic Effects of Immune Checkpoint Inhibitor Therapies

**DOI:** 10.3390/vaccines10040540

**Published:** 2022-03-31

**Authors:** Daniel Ronen, Aseel Bsoul, Michal Lotem, Suzan Abedat, Merav Yarkoni, Offer Amir, Rabea Asleh

**Affiliations:** 1Department of Internal Medicine D, Hadassah Medical Center, Faculty of Medicine, Hebrew University of Jerusalem, Jerusalem 9112001, Israel; danielronen16@gmail.com; 2Cardiovascular Research Center, Heart Institute, Hadassah University Medical Center, Faculty of Medicine, Hebrew University of Jerusalem, Jerusalem 9112001, Israel; aseel.bsoul@mail.huji.ac.il (A.B.); suzan.salman@mail.huji.ac.il (S.A.); oamir@hadassah.org.il (O.A.); 3Department of Oncology, Hadassah Medical Center, Faculty of Medicine, Hebrew University of Jerusalem, Jerusalem 9112001, Israel; mlotem@hadassah.org.il; 4Department of Cardiology, Heart Institute, Hadassah University Medical Center, Faculty of Medicine, Hebrew University of Jerusalem, Jerusalem 9112001, Israel; myarkoni@hadassah.org.il

**Keywords:** immune checkpoint inhibitors, cardiotoxicity, lymphocyte, immune system, mechanisms, adverse effects

## Abstract

Adaptive immune response modulation has taken a central position in cancer therapy in recent decades. Treatment with immune checkpoint inhibitors (ICIs) is now indicated in many cancer types with exceptional results. The two major inhibitory pathways involved are cytotoxic T-lymphocyte-associated protein 4 (CTLA4) and programmed cell death protein 1 (PD-1). Unfortunately, immune activation is not tumor-specific, and as a result, most patients will experience some form of adverse reaction. Most immune-related adverse events (IRAEs) involve the skin and gastrointestinal (GI) tract; however, any organ can be involved. Cardiotoxicity ranges from arrhythmias to life-threatening myocarditis with very high mortality rates. To date, most treatments of ICI cardiotoxicity include immune suppression, which is also not cardiac-specific and may result in hampering of tumor clearance. Understanding the mechanisms behind immune activation in the heart is crucial for the development of specific treatments. Histological data and other models have shown mainly CD4 and CD8 infiltration during ICI-induced cardiotoxicity. Inhibition of CTLA4 seems to result in the proliferation of more diverse T0cell populations, some of which with autoantigen recognition. Inhibition of PD-1 interaction with PD ligand 1/2 (PD-L1/PD-L2) results in release from inhibition of exhausted self-recognizing T cells. However, CTLA4, PD-1, and their ligands are expressed on a wide range of cells, indicating a much more intricate mechanism. This is further complicated by the identification of multiple co-stimulatory and co-inhibitory signals, as well as the association of myocarditis with antibody-driven myasthenia gravis and myositis IRAEs. In this review, we focus on the recent advances in unraveling the complexity of the mechanisms driving ICI cardiotoxicity and discuss novel therapeutic strategies for directly targeting specific underlying mechanisms to reduce IRAEs and improve outcomes.

## 1. Introduction

Immune evasion is a fundamental process crucial for tumor survival and growth. Tumor cells have been shown to manipulate the tumor microenvironment (TME) and to alter the expression of immune inhibitory proteins to achieve immune evasion [1]. In the past few decades, advances in our understanding of the adaptive immune system activation processes have allowed the development of a new class of therapies. Immune checkpoint inhibitors (ICIs) target inhibitory signals, allowing immune cell activation and clearance of tumor cells. These therapies have proven remarkably effective in a growing number of cancer types (melanoma, non-small cell lung cancer (NSCLC), colorectal cancer, renal cell carcinoma (RCC)) and are indicated for as many as 43% of cancer patients in the United States (US) in 2018 [2]. The hallmark of these therapies is the blockade of the cytotoxic T-lymphocyte-associated protein 4 (CTLA4) and programmed cell death protein 1 (PD-1) inhibitory pathways in CD4/CD8 lymphocytes (Table 1).

Unfortunately, ICI immune cell activation is not specific or selective to the TME. Systemic immune activation results in a very high incidence of immune-related adverse events (IRAE). Indeed, IRAEs are expected in up to 60% of treated patients [3], most commonly affecting the skin, gastrointestinal (GI) tract, thyroid, and lungs; however, any organ can be affected. Cardiovascular adverse events (CVAEs) from ICIs can manifest in multiple ways, ranging from conduction abnormalities to severe myocarditis [4,5], which is fatal in ~50% of cases [6]. The severity of these reactions, together with substantially increased use of ICIs, make understanding the processes governing IRAEs of great importance. Indeed, protocols have been devised for active surveillance of ICI-treated patients for the early detection of CVAEs to allow early treatment and reduce mortality. These include routine electrocardiogram (ECG) and biomarkers (mainly troponin and N-terminal (NT)-pro-B-type natriuretic peptide (NT-proBNP)), screening during the first weeks of treatment, followed by echocardiography and cardiac magnetic resonance imaging (CMR) in suspected cases [7]. Endomyocardial biopsy, though considered the gold standard for definitive diagnosis of myocarditis, is far from being ideal both in terms of its sensitivity and safety. Sensitivity may be as low as 60% [8] due to the patchy nature of myocarditis, a phenomenon also seen with ICI therapy [9]. The relatively low sensitivity and risks involved combined with the increasing diagnostic performance of other modalities, such as CMR, may make endomyocardial biopsy less necessary in many cases. Although troponin T/I and NT-proBNP were shown to be sensitive [10,11], they are far from being specific. This led to a search of new markers of ICI-related myocarditis, among which are increased neutrophile-to-lymphocyte ratio and decreased absolute lymphocyte count [12]. Additionally, the search for risk factors has shown associations of other IRAEs, such as myositis (odds ratio (OR) 25.5, 95% confidence interval (CI): 18.7–34.9; *p* < 0.001) and hepatitis (OR 2.9, 95% CI: 1.9–4.5; *p* < 0.001), pre-existing heart conditions (OR 1.26, 95% CI: 1.2–1.5; *p* < 0.001), and increased age with the development of cardiotoxicity [13,14]. In this review, we describe the emerging range of ICI-induced cardiotoxicities and address the recent advances in understanding the fundamental processes driving these events.

## 2. Adaptive Immune Response Modulation

The hallmark of T-cell activation is the specific interaction between T-cell antigen receptor (TCR) and major histocompatibility complex (MHC) molecules on antigen-presenting cells (APCs). However, this interaction is insufficient to elicit T-cell activation and co-stimulatory interactions are also needed. To our knowledge, the main co-stimulatory interaction is between CD28 on T cells and CD80/CD86 (B7-1, B7-1) on APCs. The binding of both TCR and CD28 creates the immune synapse necessary for activating exquisitely timed signaling cascades necessary for achieving specificity and coordinated T-cell activation (Figure 1) [15]. In addition to effector T cells, regulatory T cells have been shown to have major roles in tumor immune evasion [16,17]. Of these, the populations of CD4+, CD25+, Forkhead box protein P3+ (Foxp3) regulatory cells (Treg cells) have major roles. Foxp3 expression allows these cells to exert their inhibitory effects by multiple mechanisms—namely, sequestering free IL2 in the TME by expressing the high-affinity IL2-receptor, high-level expression of multiple immune checkpoint inhibitor (CTLA-4; PD-L1/2; TIGIT; GITR, OX-40; TIM-3) [18], and TGF-β and IL-10 production [19]. Indeed, Treg numbers and Treg/Teff ratios have been implicated in the poor prognosis of several cancer types [16].

## 3. Additional Stimulatory Molecules

Recent advances have identified multiple other co-stimulatory interactions [20], including *Glucocorticoid-induced tumor necrosis factor (TNF) receptor family-related protein (GITR),* expressed on intra-tumoral Treg cells, as well as on natural killer (NK) cells and tumor-infiltrating lymphocytes (TILs) [21,22,23]. Evidence from agonist antibody studies has shown that GITR activation results in increased numbers of effector T cells and reduced Treg cells [24]. *TNF receptor superfamily member 4* (TNFRSF4, *OX40*) is expressed on Treg and transiently on CD4 and CD8 cells [25]. Its activation results in Treg gain of pro-inflammatory traits, secreting interferon-gamma (INFγ), TNFα, and granzyme B [26]. *TNF receptor superfamily member 9* (TNFRSF9, *4-1BB*) can be induced on CD4 and CD8, NK, as well as other APCs [20,27]. It is used in CAR constructs and was shown to promote T-cell proliferation [28]. *Inducible T-cell co-stimulatory (ICOS)* is constitutively expressed on Treg cells and is induced in CD4 and CD8 cells following TCR/CD28 activation [29]. ICOS activation has been shown to stimulate effector T cells (Figure 2) [20,30].

To date, several attempts to harness these stimulatory effectors have shown limited effects in clinical settings [25,31,32,33,34]. One reason is the use of antibodies as agonists, resulting in antibody-dependent cellular cytotoxicity (ADCC) of target cells. The use of low-affinity antibodies or the development of small molecule agonists may prove more beneficial in the future [35]. However, current high-affinity antibodies may still be of use by driving ADCC removal of inhibitory Treg cells.

Opposing the stimulatory interaction is a set of inhibitory molecules that are believed to be important for cessation of the immune response and avoidance of self-recognition. Many inhibitory receptors were found to be upregulated subsequent to T-cell activation. This allows fast activation, followed by a shift toward a more balanced response and finally rapid cessation of the immune response after antigen removal. This is further emphasized by the exquisite timing and intensity-sensitive intracellular signaling orchestrating T-cell activation [15]. Indeed, chronic T-cell activation can result in reduced activation over time, i.e., “exhaustion” [36,37,38,39], making the inhibitory signals important for self-avoidance as well as proper timely response.

CTLA-4 and PD-1, discovered by the Nobel Prize laureates James P. Allison and Tasuku Honjo, are two important inhibitory pathways. However, several other interactions have been revealed. These molecules are expressed on different cell types, allowing multiple mechanisms for immune modulation.

## 4. CTLA4 Pathway

CTLA4 is present in intracellular vesicles in naïve T cells and is translocated to the cell membrane upon activation [15]. CTLA4 also competes with CD28 for B7-1/2 binding albeit with greater affinity [40]. In CD8 cells, CTLA4 binding results in the recruitment and activation of an SH2 domain-containing tyrosine phosphatase SYP and PP2A phosphatase, which remove TCR phosphorylation and deactivate PIK3K signaling needed for CD28 function [41,42]. This results in reduced production of IL-2 and INFγ [43,44,45] and increases the exhausted CD8 population [40]. In CD4 cells, CTLA4 activation drives differentiation into inhibitory Treg (FOXP3) rather than excitatory Th1 cells expressing inducible costimulatory molecule (ICOS) [46,47,48]. Indeed, CTLA4 is constitutively expressed on Treg cells participating in their ability to suppress immune activation and tolerance [49,50]. In B cells, CTLA4-induced immune suppression occurs predominantly via intrinsic STAT3 activation, and CTLA4 is critical for B-cell lymphoma proliferation and survival. Consequently, CTLA4 can exert its suppressive effect by utilizing multiple steps in the immune cascade, resulting in a shift toward inhibitory Treg cells rather than activating Th1 cells and reduced numbers/clones of effector CD8, NK, and B cells.

## 5. PD-1 Pathway

PD-1 receptor is an inhibitory molecule found to be upregulated following T-cell activation [51]. In turn, PD-1 binds PD ligand 1 (PD-L1) or PD-L2, which are expressed on multiple cell types, both immune and non-immune (including tumor cells) [52,53,54]. PD-1 exerts its inhibitory effect by activating two particular SH2-containing protein tyrosine phosphatases: SHP-1 and SHP-2 [55]. Similar to CTLA4 signaling, SHP-2 can dephosphorylate and thus block CD28 signaling [56] as well as the ERK and AKT pathways [57,58]. PD-1 activation also results in reduced IL-2, INFγ secretion [59], as well as inhibiting cell cycle mediators [60,61,62].

In B cells, PD-1 activation results in inhibition of B-cell antigen receptor (BCR) signaling by recruiting SHP-2 to its phosphotyrosine and dephosphorylating key signal transducers of BCR signaling [55,63], including reduced IL-6 and antibody secretion [64]. In Treg cells, PD1-PD-L1 interaction results in increased inhibitory effects on CD8 cells. Currently, the expression and the exact role of PD-1 signaling in NK-cell activation are still under investigation [51,65,66]. Taken together, these data provide strong evidence that PD-1 has a major role predominantly in T-cell inhibition via blocking the CD28 and TCR signaling pathways, resulting in anergy or exhaustion of T cells (Figure 2).

## 6. Additional Inhibitory Checkpoint Molecules

Following the discovery of CTLA4 and PD-1, additional inhibitory signals were identified: *Lymphocyte activation gene 3 (LAG3)* was found on activated T cells, B cells, and NK cells [51] and competes with CD4 for MHC-II binding with greater affinity [65,67]. In CD4 cells, LAG3 activation reduces IL-2, IL-7, IL-12, and INFγ secretion [67,68], and evidence shows its possible inhibitory effect on CD8 cells [69]. LAG3 can be cleaved by metalloproteases, resulting in an inactive soluble form (sLAG-3), which may be part of the mechanism driven by PD-1 blockade [70]. *T-cell immunoglobulin and mucin-domain containing-3 (TIM-3)* was shown to reduce CD8 response [71,72,73] and may be part of the exhausted state of T cells [74]. Additionally, it may also play a suppressive role in Th1 and Th17 cell ability to mount a response [75,76]. In contrast to these inhibitory effects, TIM-3 may also have activation effects [77,78], but its exact stimulatory role needs further investigation. *T Cell immunoglobulin and ITIM domain (TIGIT)* exerts its effect mainly on Treg, memory T cells, and NK cells [79]. TIGIT blocks the positive stimulator DNAM-1 signaling by having a higher affinity to CD155 ligand; however, it also harbors its own negative signaling, resulting in reduced IL-12 and increased IL-10 secretion [80,81,82,83]. *V-domain Ig suppressor of T-cell activation (VISTA)* is a PD-1 homolog able to exert both inhibitory and excitatory effects [84,85], mainly on T cells, macrophages, and neutrophils [86]. Initial evidence suggests that VISTA blockade results in increased T-cell activation [87]. *B7 Homolog 3 (B7H3 or CD276)* expression correlates with inhibition of T-cell function and proliferation [88,89,90]. It is expressed on T cells, NK cells, and other immune cells. Although its binding partner and mode of action have not yet been identified, initial data suggest the involvement of CD8 and NK cells [91]. *B- and T-lymphocyte attenuator (BTLA or CD272)* binding to herpes virus entry mediator (HVEM) results in SHP-2 phosphatase recruitment, similar to PD-1 signaling [92,93,94]. There is growing evidence suggesting its important role in suppressing autoimmunity [95], thus making it a promising therapeutic candidate [70,96] (Figure 2).

The discovery of multiple co-stimulatory and co-inhibitory molecules emphasizes how tightly controlled the immune response needs to be in order to achieve maximal antigen clearance with minimal collateral damage. Of the various co-inhibitory signals, CTLA4 and PD-1 are best understood with current therapies focusing on their blockage. Multiple clinical trials utilizing other co-inhibitory molecules in combination with PD-1 or CTLA4 inhibitors are underway showing promising preliminary results [25,31,32,33,34,81,97,98,99].

## 7. Innate and Antibody-Directed Tumor-Immune Modulation

As delineated above, some of the immune-modulatory molecules are expressed on a variety of immune cells, including B, NK, dendritic and myeloid cells. Apart from T-cell expression, the upregulation of co-inhibitory molecules, such as PD-1/CTLA4 and others, on APCs plays a crucial role in tumor immune avoidance. Additionally, CD73 expression on myeloid cells results in T-cell inhibition and seems independent of PD-1 blockade [100]. Furthermore, M2 macrophages may also contribute to tumor immune escape [92]. Further investigation of the effects of modulating these signals is necessary to develop additional immune-modulating therapies [101,102,103].

## 8. Mechanisms Driving IRAEs

The understanding that tumor cells utilize co-inhibitory signals to evade immune clearance drove the effort to create anti-PD-1 and Anti-CTLA4 therapies. Data show that CTLA4 blockade results in both increased CD4 cell number in the TME and a shift towards Th1 active cell type rather than the suppressive Treg cell type [46,47,48,104]. Additionally, an increased CD8 clone diversity has been observed [1,105,106,107,108,109]. On the other hand, PD-1 blockade seems to increase CD4/CD8 activity in the TME [110], probably by inhibiting entry into the exhaustion of CD8 cells or reactivating already exhausted cells [111]. Treg cells play crucial roles in TME immune suppression and show high expression levels of CTLA-4/PD1 and other co-modulatory molecules. Evidence suggests that anti-CTLA4 immune activation is dependent on ADCC removal of Treg cells [47,112,113,114]. PD1 signaling was found to promote Treg inhibitory function [115], and that loss of PD1 signaling results in Treg cells shifting to a pro-inflammatory state [116,117]. As loss of Foxp3 Treg cells results in autoimmune phenotypes in animal models, and in humans [118,119], CTLA4/PD1 inhibition may drive IRAEs by systemic inhibition of Treg cells.

Collectively, CTLA4 and PD-1 blockade seems to exert major effects primarily on T-cell immunity. Indeed, studies have demonstrated that the amount of T lymphocytes in the TME is correlated with ICI treatment [120]. However, the wide range of cells expressing co-inhibitory molecules indicates that there are multiple mechanisms involved. Evidence suggests antibody diversity of both antitumor and autoantibodies are increased [121] following ICI blockade and that the antibody levels are correlated with the IRAE severity [122,123]. Furthermore, recent findings are supportive of specific antibodies implicated in ICI hypophysitis, pneumonitis, diabetes mellitus, and hypothyroidism [124,125,126,127]. The association between ICI myocarditis and myositis and myasthenia gravis [5,6] further supports the role of antibodies in IRAE, including in the heart. Additional cell types, such as neutrophils, have been shown to be activated during CTAL4 blockade [128,129]. Following ICI treatment monocytes were found to be recruited to the pancreas and liver by increased INFγ, resulting in islet cell damage [130,131], and M2 macrophages were found in autopsies of IRAE patients [132]. Elevated cytokine production identified in concordance with an activated immune response may drive further damage, similar to that seen in autoinflammatory diseases. Indeed, IL-1 levels were found to be elevated in patients suffering from IRAEs [133,134]. Additionally, TNFα, IL-17, INFγ, and IL-6 are all involved in autoimmune reactions and were shown to be in correlation with IRAEs or overexpressed following PD-1 inhibition [105,135]. Finally, the expression of inhibitory or excitatory molecules on non-immune cell types may drive cell-type-specific IRAEs, as seen with CTLA4 expression on hypothalamic and pituitary cells [136].

Based on this understanding, immune checkpoint inhibition damage in non-tumor sites can result from several mechanisms (Figure 3):Increased T-cell clone expansion results in the appearance of new self-recognizing effector cells [4,137,138,139];The release from inhibition of preexisting self-recognizing effector T cells [140];Tumor damage exposure of autoantigens, resulting in expansion of auto-recognizing clones [4,141,142,143,144];Antibody-mediated autoimmunity [124,125];A shift toward a more active and less selective innate immune response;Bystander damage from increased levels of cytokines produced by the active antitumor immune response [133,134];Direct damage resulting from specific molecule expression on target cells.

Ironically, the appearance of IRAEs seems to corollate with better antitumor response [145,146,147,148,149,150,151,152], making ICIs a double-edged sword with a need for balancing the antitumor effect against the severity of IRAEs.

## 9. Cardiotoxicity of ICIs

The increased use of ICI therapy has revealed the extent and importance of ICI-induced cardiotoxicity (Figure 4). The severity of cardiac IRAEs is classified according to the American Society of Clinical Oncology [153] and ranges from benign ECG changes to life-threatening myocarditis. Recently, analysis of the Pharmacovigilance database and a meta-analysis of published papers and randomized controlled trials (RCTs) has identified cardiac events in 1.3–5.8% of patients receiving ICIs (single and dual therapy) [154]. Another large meta-analysis of RCTs showed increased odds ratios for developing myocarditis and pericarditis (4.4 and 2.2, respectively) in ICI-treated patients, with increased associated risks of heart failure and myocardial infarction (MI) [155]. Surprisingly, a meta-analysis of RCTs from phase II and III trials did not find any difference in CVAE between ICI-treated and control groups [156]. This may be explained by including highly selected patients in these trials (selection bias), reporting bias, and the type of analysis [155].

The immune landscape of the resting heart is mostly dominated by macrophages [157,158], with low numbers of monocytes, dendritic cells, and mast cells present [159,160]. In addition, low numbers of T and B cells can be found [161,162]. Myocarditis, inflammation of the heart muscle, can be triggered by multiple etiologies and is characterized by lymphocyte-predominant infiltration (such as in viral myocarditis), eosinophilic infiltration (such as in hypereosinophilic syndrome), or the presence of giant cells (giant cell myocarditis). Like other IRAEs, ICI cardiotoxicity is also driven by lymphocyte, predominant immunity with CTLA4 and PD-L1 expression in the myocardium playing an important role in T-cell activity regulation [163,164,165,166].

## 10. Myocarditis

The most serious cardiac manifestation of ICI cardiotoxicity is the development of myocarditis. Although it may be silent with merely cardiac biomarker elevation, it may present as a fulminant disease, leading to cardiogenic shock and death [11]. A meta-analysis has reported that myocarditis may represent up to 50% of cardiac IRAEs (34 of 61 cardiac IRAEs among 4751 patients) [154], with mortality rates ranging from 20% to 50% [5,6,10,11,14]. Cardiac damage seems to start early after ICI treatment, with myocarditis reported to occur approximately 17 days (range: 13–64 days) following treatment initiation [4], but delayed appearance as far as 32 weeks after treatment initiation was reported [167]. PD-1 pathway inhibition seems to elicit more cardiotoxicity than CTLA4, and combinational therapy harbors the highest risk of myocarditis [4,10,168]. Interestingly, female patients are more likely to develop IRAEs and myocarditis [169,170]. This may be explained, at least partially, by PD-1 and PD-1L upregulation by estrogen [171,172], and is supported by an observation from CTLA4(+/−)/PD-1(−/−) deficient mice showing that female mice died at significantly higher rates than their male, age-matched counterparts [173].

Blockade of PD-1 and CTLA4 was shown to result in increased lymphocyte presence in the heart and other tissues [174]. This is supported by histological data from ICI-induced myocarditis patients showing increased CD4 and CD8 lymphocytes and, to a lesser degree, macrophages [4,175,176]. Interestingly, these T cells seem to have lower stimulatory thresholds for self-antigens [139]. Mouse models have shown that PD-1L is upregulated in endothelial and myocardial cells in response to inflammation or ischemia, presumably reducing the inflammatory response and tissue damage [4,177,178,179]. Several approaches have been perused to elucidate the mechanisms driving myocardial damage during ICI treatment. Initial mouse models showed varied effects, depending on the genetic background. PD-1 knockout in C57BL/6 mice displayed a normal phenotype [180], whereas deleting PD-1 in BALB/c background resulted in increased anti-troponin-I antibodies and dilated cardiomyopathy rather than myocarditis [143,163,180,181]. However, the transfer of CD8+ PD-1-deficient cell population to mice already experiencing myocarditis resulted in enhanced inflammation [182]. In MRL-lpr−/− mice (lacking FAS and predisposed to the development of systemic lupus erythematosus (SLE)-like phenotype), deletion of PD-1/PD-1L or the administration of neutralizing antibodies resulted in autoimmune myocarditis and CD4/8 infiltration; however, unlike in human ICI myocarditis, substantial levels of autoantibody formation against myosin were detected [173,183]. Interestingly, blocking PD-1 or CTLA4 in MRL mice results in increased lymphocyte infiltration without the development of overt myocarditis [173]. These differences between early mouse models and human ICI myocarditis may be explained by the fact that in mice most T cells are naïve, as opposed to most human T cells [173]. This notion is supported by studies examining myocarditis developing in melanoma mouse models treated with PD-1 blockade [166]. In contrast to solitary PD-1 deficiency, mouse models of CTLA-4 deficiency led to systemic inflammation involving T lymphocytes and resulted in early multiorgan failure and death [164,165]. This included myocarditis with CD8 cells infiltration [164,184]; however, the global activation of T lymphocytes also differs from ICI myocarditis. In light of these shortcomings of early mouse models, a primate model was developed showing CD4/CD8 T cell and low numbers of macrophage infiltration into cardiac tissues, with increased troponin-I and NT-proBNP levels [175]. Another approach to recapitulate ICI myocarditis in mice employed removing multiple participating genes. Deletion of both PD-1 and LAG-3 in BALB/c mice resulted in myocarditis with CD8 and CD4 cell infiltration, in addition to increased TNFα secretion [185,186]. More recently, a mouse model harboring a compound loss of CTLA4 and PD-1 was created. Ctla4+/− Pdcd1−/− mice showed increased troponin levels with cardiac infiltration of CD3, CD8, and CD4 cells, and reduced numbers of Treg cells, similar to that seen in human ICI myocarditis [173]. A more “physiologic” approach was employed by Lars et al., who injected immune-competent mice with melanoma cells, followed by anti-PD-1 antibodies treatment, resulting in functionally noticeable myocarditis with increased CD4 and CD8 cell infiltration [166]. Both Ctla4+/− Pdcd1−/− and melanoma models were used to assess treatment options (CTLA4-IgG1 and anti-TNFα, respectively) paving the way for a more comprehensive search for novel and more specific therapeutic options.

One reason for T-cell infiltration within the myocardium was elucidated by the recognition of similar clonal populations of T cells recognizing antigens expressed in both the TME and cardiac tissues [4]. Shared antigens between myeloma cells and cardiomyocytes have been later discovered [144]. This mechanism is further supported by the finding of similar TME and skin T cells in NSCLC presenting with skin IRAEs [142], suggesting a common underlying IRAE mechanism. In addition to antigen recognition, ICI treatment results in upregulation of CXCR3–CXCL9/CXCL10 and CCR5/CCL5 chemokines necessary for T-cell activation [175,187]. Following activation, T cells overexpress TNFα, INFγ, and granzyme B, thus promoting local cell damage and death [188,189,190]. This is also supported by in vitro co-culture experiments of cardiomyocytes exposed to lymphocytes and anti-PD-1/CTLA4 antibodies, demonstrating increased levels of leukotriene B4 [188]. The surprisingly high incidence of myasthenia gravis with ICI myocarditis suggests that, in addition to T cells, antibody-mediated damage might also contribute to ICI autoimmunity [191,192,193,194,195]. Antibodies against AChR and MuSK are commonly present (~66% (30 of 45 patients) and 5% (1 of 19 patients), respectively) in ICI-induced myasthenia gravis cases [196]. However, as mentioned, increased anti-troponin-I and anti-myosin antibodies seen in PD-1 mouse models result in dilated cardiomyopathy or myocarditis that differs from ICI myocarditis. Further investigation is needed to elucidate this aspect.

Taken together, myocarditis seems to develop by multiple steps including enhanced recruitment of T lymphocytes, reduced PD-1/PD-1L protection, reduced Treg activity, and T-cell recognition of shared epitopes. The development of novel mouse models will potentially allow better elucidation of other mechanisms driving ICI myocarditis and drive better-targeted treatment options (Figure 3).

## 11. Pericarditis

Pericardial effusion was reported following ICI treatment, with some developing overt pericarditis [197,198]. The incidence of pericardial effusion ranges from 0.3% (95/31321 cardiovascular IRAEs) but may be as high as 7% in NSCLC (4/60 patients receiving ICIs), and this has been associated with worse outcomes [6,199]. As pericardial effusion is present in multiple tumor types, it is important to differentiate the etiology driving pericardial effusion accumulation; however, the association of pericardial effusion with higher mortality in ICI-treated patients suggests a possible cardiotoxic effect. Indeed, pericardial samples from ICI-treated patients suffering from pericarditis showed significant lymphocytic infiltrates [200].

## 12. Arrhythmias

Atrial fibrillation (AF), conduction delays, and ventricular arrhythmias have all been observed following ICI treatment [4,11,139,201], and cardiac arrhythmias may represent a common cardiac IRAE [6,11,202,203] that may occur in up to 10% of patients receiving ICIs, according to one study [202]. This may represent isolated conduction toxicity or be part of a widespread manifestation of myocarditis [11]. Interestingly, CD4 PD-1 expression and PD-1L myeloid dendritic cells were found to be reduced in patients with AF [204]. Additionality, mouse models of myocarditis showed an increased incidence of arrhythmias [173]. The significantly worse outcome of patients presenting with conduction abnormalities (80% vs. 16% mortality) [11] following ICI treatment emphasizes its significance and the importance of ECG surveillance.

## 13. Atherosclerosis

Atherosclerosis has emerged as a new target of ICI-induced toxicity [205]. Indeed, acute coronary syndrome (ACS) incidence among ICI patients was reported to be around 0.5% (2/402 of ICI-treated patients) in a large metanalysis [206], but other reports of incidences as high as 2.4% (80/3326) to 3.6% (102/2842) of patients receiving ICI have emerged [14,207,208]. Although macrophages play major roles in the development of the atherosclerotic lesion, T lymphocytes were shown to occupy the atherosclerotic plaque [209] ], and their activation and secretion of INFγ lead to macrophage activation [210]. Interestingly, patients with coronary artery disease have reduced expression of PD-1 and PD-L1 on peripheral mononuclear cells [211]. The role of PD-1 in atherosclerosis is further supported by observations of larger atherosclerotic lesions in PD-1 knockout mice [212] with a more prominent T-cell response and larger necrotic cores [213]. As for CTLA-4, studies in mice have shown increased plaque size, following anti-CTLA-4 antibody treatment. Concurrently, treatment with abatacept (a CLTA-4 analog) or overexpression of CTLA-4 reduced plaque size [214,215,216]. The proatherosclerotic effect of PD-1 and CTLA4 inhibition seems to be, at least in part, mediated by the increase in INFγ and TNFα secretion, which are known mediators promoting coronary thrombosis [217]. As new immunomodulatory targets are discovered, it is interesting to see whether it can be possible to activate the immune response without increasing atherosclerotic risk [205].

Events of coronary spasm following ICI administration have been reported [218,219], but the full extent of this phenomenon warrants further investigation.

## 14. Hypertension

In addition to direct cardiotoxicity, ICI-related hypertension may add to cardiac morbidity and mortality. Hypertension seems to be a rear IRAE and was not reported in phase III trials [220,221,222]. However, more recent data show that hypertension may develop in as high as 18% (13/70) of treated patients and is reported to account for 0.63% of ICI-related IRAEs [6,223].

## 15. Thromboembolism

Venous thromboembolism (VTE), although indirect, has a profound effect on the cardiovascular system. As cancer and certain chemotherapies are known to induce a pro-thrombotic state, and as ICIs are commonly used as second-line therapies, it is difficult to assess the isolated contribution of ICI treatment to thromboembolism formation. Initial data, including RCT and metanalysis data, showed that approximately 2.7% (449/20273 ICI-treated patients) developed VTE during ICI therapy [224,225,226]. However, this analysis included studies in which VTE was not the primary outcome, nor was standardly reported, resulting in possible underrepresentation. New emerging data show an increased incidence of pulmonary embolism (PE) and deep vein thrombosis (DVT) [227], suggesting the prothrombotic effects of ICIs. Indeed, multiple retrospective studies have shown a much higher VTE incidence of 6–24% in ICI-treated patients, with risk increasing after prolonged treatment [228,229,230,231,232]. Furthermore, these studies have found that the main risk factors for VTE in ICI-treated patients include the presence of metastatic disease and previous VTE and that these risk factors are also associated with a worse prognosis. Interestingly, no correlation was found in these studies between ICI type or cancer type and VTE risk. Despite a potentially increased risk of VTE, it is worth noting that these studies were all retrospective and did not include non-ICI-treated control groups. Furthermore, several studies have shown that VTE risk was similar between patients treated with ICIs and those treated with chemotherapy [225,233].

Mechanistically, ICIs may induce VTE by causing vasculitis and endothelial damage secondary to local and systemic activation of immune cells [228]. A more specific role of PD1/PD-L1 signaling has been suggested in one study showing that T-cell activation in ICI-treated patients induces production of tissue factor in peripheral monocytes highly expressing PD-L1, which can initiate the thrombotic cascade [234]. Taken together, it seems VTE is emerging as an important IRAE. However, a better understanding of the underlying mechanisms, as well as controlled clinical trials, is necessary to explore the extent and risk factors of VTE associated with ICI use.

## 16. Emerging Treatment of Cardiac IRAEs

To date, the hallmark of IRAE management is based on steroid therapy and escalation to other immunosuppressive regiments in non-responders [7,153]. Immune suppression using the inosine-5′-monophosphate dehydrogenase (IMPDH) inhibitor, mycophenolate mofetil (MMF), results primarily in B- and T-cell inhibition [235,236]. Calcineurin inhibitors, such as tacrolimus, exert immunosuppressive properties through inhibiting T-cell activity [167,237,238,239]. Lymphopenia may be induced by antithymocyte globulin (ATG) [167,237,238], resulting in the cessation of autoimmunity. Other strategies include the use of intravenous immunoglobulin (IVIG) [191,192,193,240,241,242,243], and plasmapheresis [194,195,240,244,245,246] to incapacitate or remove ICIs (which have a long half-life of up to 27 days in some cases [201] can be employed.

Recent advances in the understanding of IRAE mechanisms have allowed the employment of targeted therapies. These include CTLA4-Ig (abatacept) to reimpose inhibition of activated T cells in myocarditis, as shown in mouse models [173] and case reports in humans [173,247]. Anti-CD52 (alemtuzumab), commonly used in CLL and multiple myeloma, has shown T-cell depletion capabilities and has been used in a case report with encouraging results [248]. Anti-TNFα (infliximab) has also shown promising results [153,166,240,249], although its utility in heart failure patients may be limited. It is worth noting that any treatment resulting in global quenching of the immune response also has the potential to allow tumor escape from immunosurveillance and disease progression. A better understanding of both the stimulatory and inhibitory molecules implicated in intracellular signaling pathways may allow better control of adverse events while allowing the tenacious antitumoral activity.

## 17. Conclusions

Harnessing the immune response to facilitate tumor clearance has opened up a new avenue in cancer treatment. The achievement of successful treatment results has made PD-1/CTLA4 inhibition available to an increasing number of cancer patients. With this increase came the realization of various adverse events accompanying this treatment type. Cardiac toxicity is emerging to be far more common, with grave consequences even in more benign manifestations. Recent advances in understanding the mechanisms behind ICI treatment have allowed new treatment options such as abatacept and anti-TNFα agents that can reduce drug toxicity and allow continuation of treatment to achieve tumor clearance. Furthermore, the discovery of new stimulatory and inhibitory molecules may allow the development of ICIs with safer adverse event profiles.

## Figures and Tables

**Figure 1 vaccines-10-00540-f001:**
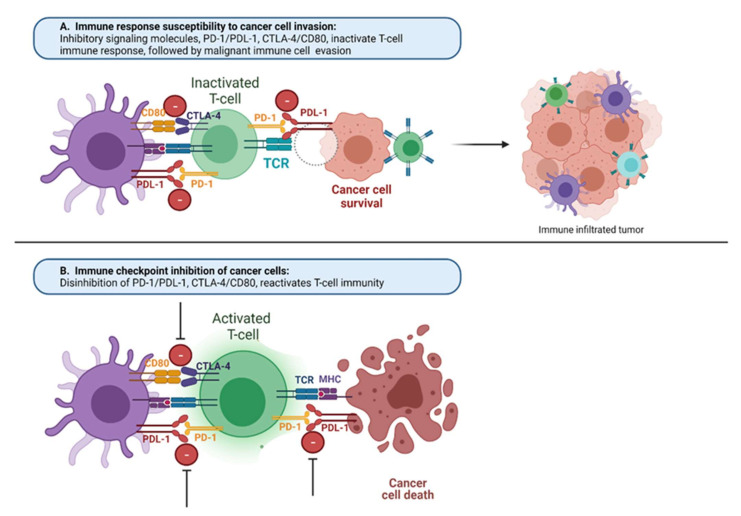
**Mechanisms of the immune response under malignancy and with immune checkpoint inhibitors:** (**A**) immune response susceptibility to cancer cell invasion. Common ligands, MHC and PDL-1, found on immune macrophages (purple) and cancer cells (brown) compete with binding to receptors on T cells (green) TCR and PD-1, which normally regulate the immune response. When immunologic homeostasis is compromised between positive (TCR/MHC) and negative (CTLA-4/CD80, PD-1/PDL-1) signaling of T-cell activation, cancer cells can avoid the immune response of the host and escape from immune cell attack, providing cancer cell growth and proliferation uncontrollably; (**B**) in presence of immune checkpoint inhibitors toward CTLA-4/CD80, PD-1/PDL-1 signaling, there is disinhibition of these pathways and reactivation of T cells toward cancer cell death. MHC, major histocompatibility complex; PD-1, programmed cell death protein 1; PD-L1, programmed death-ligand 1; CTLA-4, cytotoxic T-lymphocyte-associated protein 4; TCR, T-cell receptor.

**Figure 2 vaccines-10-00540-f002:**
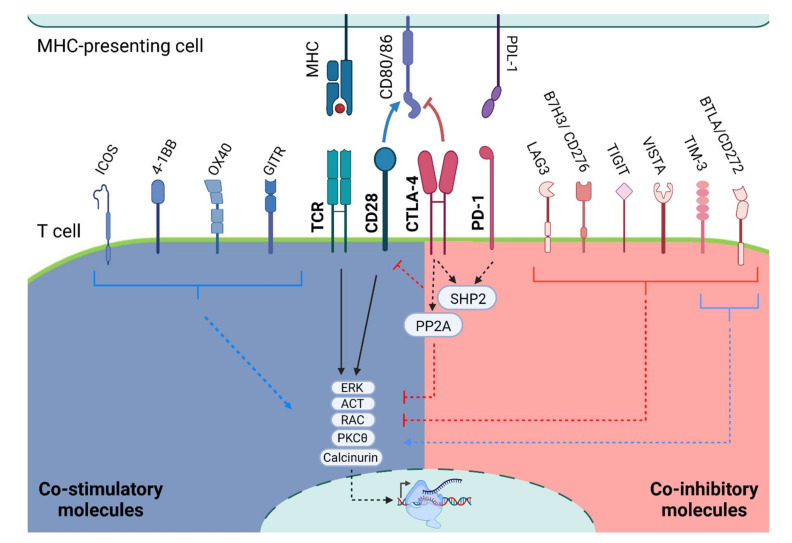
**Co-stimulatory and Co-inhibitory signaling.** Schematic representation of TCR activation with the co-stimulatory and co-inhibitory molecules. TIM-3 and BTLA have shown both inhibitory and stimulatory activity. It is worth noting that the T-cell signaling depicts a general interaction of T cells with any MHC-presenting cells and is not specific to one cell type: MHC, major histocompatibility complex; PD-1, programmed cell death protein 1; PD-L1, programmed death-ligand 1; CTLA-4, cytotoxic T-lymphocyte-associated protein 4; TCR, T-cell receptor; GITR, glucocorticoid-induced tumor necrosis factor (TNF) receptor family-related protein; OX40 is also known as TNF receptor superfamily member 4 (TNFRSF4); 4-1BB is also known as TNF receptor superfamily member 9 (TNFRSF9); ICOS, inducible T cell co-stimulatory; LAG3, lymphocyte activation gene 3; TIM-3, T-cell immunoglobulin, and mucin-domain containing-3; TIGIT, T-Cell immunoglobulin and ITIM domain; VISTA, V-domain Ig suppressor of T-cell activation; B7H3 or CD276, B7 homolog 3; BTLA or CD272, B- and T-lymphocyte attenuator.

**Figure 3 vaccines-10-00540-f003:**
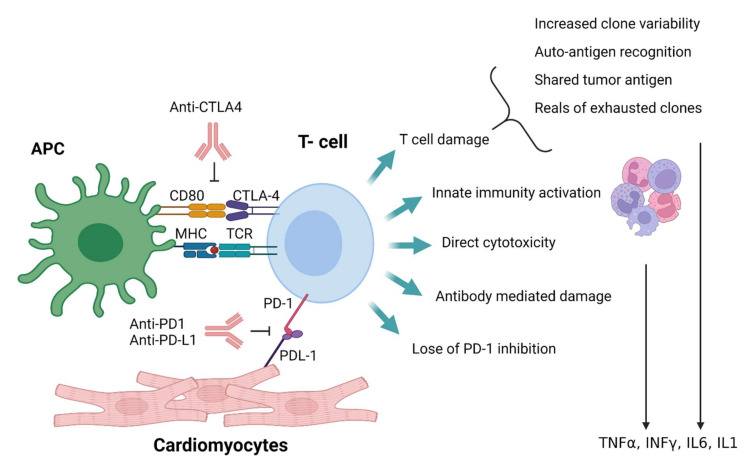
**Depiction of possible IRAE mechanisms:** APC, antigen-presenting cell; MHC, major histocompatibility complex; PD-1, programmed cell death protein 1; PD-L1, programmed death-ligand 1; CTLA-4, cytotoxic T-lymphocyte-associated protein 4; TCR, T-cell receptor; TNFα, tumor necrosis factor-alpha; INFγ, interferon-gamma; IL1, interleukin 1; IL6, interleukin 6.

**Figure 4 vaccines-10-00540-f004:**
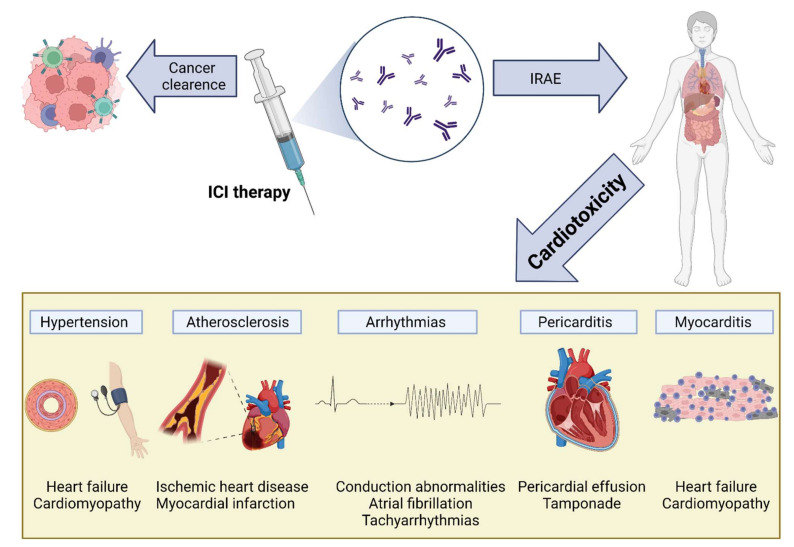
**Cardiovascular toxicity induced by immune checkpoint inhibitor (ICI) therapy.** The reactivation of T cells in response to ICIs is mediated by inflammation and fibrosis, leading to various cardiac manifestations, including myocarditis, pericarditis and pericardial effusion, arrhythmias, atherosclerosis, and hypertension: IRAEs, immune-related adverse events; ICI, immune checkpoint inhibitor.

**Table 1 vaccines-10-00540-t001:** List of available immune checkpoint inhibitors licensed by the FDA.

Mechanism of Action	Drug
Anti-CTLA4	Ipilimumab
	Tremelimumab
Anti-PD-1	Nivolumab
	Pembrolizumab
	Cemiplimab
Anti-PD-L1	Atezolizumab
	Avelumab
	Durvalumab

Abbreviations: FDA, Food and Drug Administration; CTLA4, cytotoxic T-lymphocyte-associated protein 4; PD-1, programmed cell death protein 1; PD-L1, programmed cell death protein ligand 1.

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
