# Peer review of "Exploring the Mechanisms Underlying the Cardiotoxic Effects of Immune Checkpoint Inhibitor Therapies"

_vaccines, 2022, doi:10.3390/vaccines10040540_

Round 1

Reviewer 1 Report

Treatment with Immune checkpoint inhibitors (ICI), including cytotoxic T-lymphocyte-associated protein 4 (CTLA4) and programmed cell death protein 1 (PD-1), is currently an emerging cancer treatment with promising results. Their use is related to the development of adverse events.

In this review, the authors addressed cardiotoxicity.

They discuss available data on histological features and T cell populations. They focus on the recent advances in unraveling the complexity of the mechanisms driving ICI cardiotoxicity and discuss novel therapeutic strategies for directly targeting specific underlying mechanisms to reduce immuno-related AEs and improve outcomes.

The manuscript is of interest and of clinical relevance due to the growing use of immunotherapies in different oncological settings.

However, some points deserve further data and should be addressed.

-As the development of immuno-related AEs is the main limiting factor of immunotherapy, the authors would further discuss the potential mechanisms leading to the development of adverse events. In particular, when they discuss "Mechanisms driving IRAEs" they should recall the available data regarding the role of CD4+ CD25 Foxp3 Regulatory T cells (Treg) in the development of immuno-related adverse events. As Tregs amply express checkpoint molecules such as cytotoxic T lymphocyte-associated antigen 4 and programmed cell-death 1 receptor, they represent a direct target of immune checkpoint inhibitor (ICI) immunotherapy. Taking into consideration the critical role of Tregs in the maintenance of immune homeostasis as well as avoidance of autoimmunity, it has been reported that targeting of Tregs by ICI immunotherapy may result in the development of immune-related adverse events, as recently well described (Hepatocellular carcinoma in viral and autoimmune liver diseases: Role of CD4+ CD25+ Foxp3+ regulatory T cells in the immune microenvironment. World J Gastroenterol. 2021 Jun 14;27(22):2994-3009).

Author Response

We thank the Reviewer for these positive and insightful comments. We have addressed all issues raised by the reviewer and point-by-point responses to their comments are provided here.  

Reviewer #1:

Treatment with Immune checkpoint inhibitors (ICI), including cytotoxic T-lymphocyte-associated protein 4 (CTLA4) and programmed cell death protein 1 (PD-1), is currently an emerging cancer treatment with promising results. Their use is related to the development of adverse events.

In this review, the authors addressed cardiotoxicity.

They discuss available data on histological features and T cell populations. They focus on the recent advances in unraveling the complexity of the mechanisms driving ICI cardiotoxicity and discuss novel therapeutic strategies for directly targeting specific underlying mechanisms to reduce immuno-related AEs and improve outcomes.

The manuscript is of interest and of clinical relevance due to the growing use of immunotherapies in different oncological settings.

However, some points deserve further data and should be addressed.

Comment: As the development of immuno-related AEs is the main limiting factor of immunotherapy, the authors would further discuss the potential mechanisms leading to the development of adverse events. In particular, when they discuss "Mechanisms driving IRAEs" they should recall the available data regarding the role of CD4+ CD25 Foxp3 Regulatory T cells (Treg) in the development of immuno-related adverse events. As Tregs amply express checkpoint molecules such as cytotoxic T lymphocyte-associated antigen 4 and programmed cell-death 1 receptor, they represent a direct target of immune checkpoint inhibitor (ICI) immunotherapy. Taking into consideration the critical role of Tregs in the maintenance of immune homeostasis as well as avoidance of autoimmunity, it has been reported that targeting of Tregs by ICI immunotherapy may result in the development of immune-related adverse events, as recently well described (Hepatocellular carcinoma in viral and autoimmune liver diseases: Role of CD4+ CD25+ Foxp3+ regulatory T cells in the immune microenvironment. World J Gastroenterol. 2021 Jun 14;27(22):2994-3009).

RE: We thank the Reviewer for raising this issue. We did mention the role of Treg in the TME, however we agree with the comment that further elaboration was needed to emphasize the importance of these cell type in ICI biology. This was done by further expanding sections on " Adaptive immune response modulation " (page 3) and on " Mechanisms driving IRAE" (Page 7). We have also added the suggested as well as other relevant references.

Reviewer 2 Report

This paper is a review of adverse events involving the cardiovascular system after immune check point cancer therapy.  In general, the paper is informative and interesting.  However, it needs improvement. First up, there are no numbers for the references and no line numbers for reviewers to note issues with paper.  There are a few text errors.  Cardiovascular adverse events evolve from CVAEs to CAVEs after first mention.  

Page 5 CTLA4 section, line 5-singling , signaling

Page 6 Innate and antibody--- section, line 5 blocked, blockade

Page 7 Mechanisms driving IRAEs section, line 4 autoantibodies is, are

Line 12 M2 macrophage were, macrophages were

Probably a few more.  Please, do some more checking.

I find the percentages given in the text a little confusing. This begins on page 2 after Table 1, where CVAEs occur in around 1-5% of patients-----fatal in 50% of cases.  This is better explained later in the paper on page 9 under Myocarditis.  The early mention could just raise the issue of fatal events and leave the percentage issue to the myocarditis section.  Also, we have mention of percentages on each of the subsections, 7% for pericarditis, 10% arrhythmias, 7% atherosclerosis, 0.63% hypertension, 2.7% thromboembolism.  % of what?; total ICI or % of the 1-5% CVAEs or % of the 60% IRAEs.  Also, can we see some actual numbers.  How many patients are represented in the CVAE population (a rough estimate would do)?

On page 2, last 3 lines, a mention of prognostic indicators for CVAEs is presented. What is the relative risk (HR 95% CI, p value for these 2 indicators, age and pre-existing heart condition?  Do these factors ever dissuade the chose to use ICI?

On page 12 in the Emerging treatment section, please spell out how each of the IRAE management therapies work other than steroids,IVIG and plasmapheresis and how these might or might not play into allowing the cancer to progress.

Author Response

We thank the Reviewer for these positive and insightful comments. We have addressed all issues raised by the reviewer and point-by-point responses to their comments are provided here.  

Comment: This paper is a review of adverse events involving the cardiovascular system after immune check point cancer therapy.  In general, the paper is informative and interesting.  However, it needs improvement. First up, there are no numbers for the references and no line numbers for reviewers to note issues with paper.  There are a few text errors.  Cardiovascular adverse events evolve from CVAEs to CAVEs after first mention.  

RE: We appreciate the Reviewer`s comments. We have added numbers to the references and corrected the other typos. 

Comment: Page 5 CTLA4 section, line 5-singling, signaling

RE: Corrected.

Comment: Page 6 Innate and antibody--- section, line 5 blocked, blockade

RE: Corrected.

Comment: Page 7 Mechanisms driving IRAEs section, line 4 autoantibodies is, are

RE: Corrected.

Comment: Line 12 M2 macrophage were, macrophages were

RE: Corrected.

Comment: I find the percentages given in the text a little confusing. This begins on page 2 after Table 1, where CVAEs occur in around 1-5% of patients-----fatal in 50% of cases.  This is better explained later in the paper on page 9 under Myocarditis.  The early mention could just raise the issue of fatal events and leave the percentage issue to the myocarditis section.  Also, we have mention of percentages on each of the subsections, 7% for pericarditis, 10% arrhythmias, 7% atherosclerosis, 0.63% hypertension, 2.7% thromboembolism.  % of what?; total ICI or % of the 1-5% CVAEs or % of the 60% IRAEs.  Also, can we see some actual numbers.  How many patients are represented in the CVAE population (a rough estimate would do)?

RE: We thank the reviewer for this valuable comment. Indeed, the percentages given in the text needed further explanation. We have added the relative information and included the actual patient numbers where appropriate, as requested.

Comment: On page 2, last 3 lines, a mention of prognostic indicators for CVAEs is presented. What is the relative risk (HR 95% CI, p value for these 2 indicators, age and pre-existing heart condition?  Do these factors ever dissuade the chose to use ICI?

RE: We have added the odds ratios (OR) and 95% CI for the relevant data and have further elaborated on the implications of these indicators (Page 2 last 2 lines).

Comment: On page 12, in the Emerging treatment section, please spell out how each of the IRAE management therapies work other than steroids, IVIG and plasmapheresis and how these might or might not play into allowing the cancer to progress.

RE: We appreciate the Reviewer`s comment. As requested, we have given a short explanation on the mechanisms underlying each specific therapy (page 12).

Round 2

Reviewer 2 Report

The paper may still have English usage errors.  For example, on page 13-hypertension section- Hypertension seems to be a rear (rare) IRAE-

Please check text again for any additional problems.